

# Optimal and near-optimal exponent-pairs for the Bertalanffy-Pütter growth model

Katharina Renner-Martin, Norbert Brunner, Manfred Kühleitner, Werner-Georg Nowak and Klaus Scheicher

Department of Integrative Biology and Biodiversity, Institute of Mathematics, Universität für Bodenkultur Wien, Wien, VIE, Austria

## ABSTRACT

The Bertalanffy–Pütter growth model describes mass $m$ at age $t$ by means of the differential equation $dm/dt = p * m^a - q * m^b$. The special case using the von Bertalanffy exponent-pair $a = 2/3$ and $b = 1$ is most common (it corresponds to the von Bertalanffy growth function VBGF for length in fishery literature). Fitting VBGF to size-at-age data requires the optimization of three model parameters (the constants $p$, $q$, and an initial value for the differential equation). For the general Bertalanffy–Pütter model, two more model parameters are optimized (the pair $a < b$ of non-negative exponents). While this reduces bias in growth estimates, it increases model complexity and more advanced optimization methods are needed, such as the Nelder–Mead amoeba method, interior point methods, or simulated annealing. Is the improved performance worth these efforts? For the case, where the exponent $b = 1$ remains fixed, it is known that for most fish data any exponent $a < 1$ could be used to model growth without affecting the fit to the data significantly (when the other parameters were optimized). We hypothesized that the optimization of both exponents would result in a significantly better fit of the optimal growth function to the data and we tested this conjecture for a data set (20,166 fish) about the mass-growth of Walleye (*Sander vitreus*), a fish from Lake Erie, USA. To this end, we assessed the fit on a grid of 14,281 exponent-pairs $(a, b)$ and identified the best fitting model curve on the boundary $a = b$ of the grid ($a = b = 0.686$); it corresponds to the generalized Gompertz equation $dm/dt = p * m^a - q * \ln(m) * m^a$. Using the Akaike information criterion for model selection, the answer to the conjecture was no: The von Bertalanffy exponent-pair model (but not the logistic model) remained parsimonious. However, the bias reduction attained by the optimal exponent-pair may be worth the tradeoff with complexity in some situations where predictive power is solely preferred. Therefore, we recommend the use of the Bertalanffy–Pütter model (and of its limit case, the generalized Gompertz model) in natural resources management (such as in fishery stock assessments), as it relies on careful quantitative assessments to recommend policies for sustainable resource usage.

Corresponding author
Katharina Renner-Martin, kathi.renner-martin@gmx.de

## INTRODUCTION

Size-at-age (length or mass) is an important metric about animals (Google search: ca. 286,000 results), in particular for fisheries management (*Ogle & Iserman, 2017*). Consequently, various models for size-at-age have been proposed, whereby models aiming at biological explanations for growth focus on mass-at-age (c.f. *Ursin, 1979*; *Enberg, Dunlop & Jørgensen, 2008*). Here, we investigate a general class of growth models in terms of the *Von Bertalanffy (1957)* and *Pütter (1920)* differential equation (1):

$$\frac{dm(t)}{dt} = p \cdot m(t)^a - q \cdot m(t)^b \tag{1}$$

Equation (1) describes body mass (weight) $m(t) > 0$ as a function of age $t$, using five model parameters: $a, b, p, q, m_0$. Thereby, $m_0 > 0$ is an initial value, that is, $m(0) = m_0$. The exponent-pair $a < b$ ("metabolic scaling exponents") is non-negative and also the constants $p$ and $q$ are non-negative. Several "named models" are special instances of (1): To describe mass-at-age, *Von Bertalanffy (1957)* suggested the exponent-pair $a = 2/3$ and $b = 1$, *West, Brown & Enquist (2001)* proposed $a = 3/4$, $b = 1$, other authors considered $a = 1$, $b = 2$ (logistic growth of *Verhulst, 1838*), *Richards (1959)* recommended $a = 1$ while retaining $b > 1$ as a free parameter, and the generalized Bertalanffy growth model assumes $b = 1$, using $a < 1$ as parameter (recommended, e.g., by *Pauly, 1981*). There are also models of type (1) for length-at-age, notably VBGF, the von Bertalanffy growth function with exponent-pair $a = 0$, $b = 1$ (bounded exponential growth) which is widely used in fishery literature (Google search for "VBGF, fish": ca. 15,000 results). VBGF is equivalent to the model with the von Bertalanffy exponent-pair ($a = 2/3$, $b = 1$) for mass-growth (*Von Bertalanffy, 1957*).

In the case of equal exponents, the generalized Gompertz differential equation (2) replaces Eq. (1). Its right-hand side is the limit of the right-hand side of (1), assuming $b$ approaches $a$. Its special case $a = 1$ defines the *Gompertz (1832)* model;

$$\frac{dm(t)}{dt} = p \cdot m(t)^a - q \cdot \ln(m(t)) \cdot m(t)^a \tag{2}$$

In general, the solutions of (1) and (2) involve non-elementary functions, namely hypergeometric functions and exponential integrals, respectively (*Ohnishi, Yamakawa & Akamine, 2014*; *Marusic & Bajzer, 1993*; further explanations: *Seaborn, 2013*). The solutions of the more special "named models" are elementary.

Concrete values for the parameters of Eqs. (1), (2) are obtained by identifying a growth function (i.e., a concrete solution of the differential equations) with the best fit to the data. Experience has shown that no single of the above-mentioned "named models" was exactly correct for all species (c.f. *Killen, Atkinson & Glazier, 2010* for fish; *White, 2010* for mammals). *Renner-Martin et al. (2018)* explored the situation for the generalized von Bertalanffy model (the exponent $b = 1$ is held fixed) and found that for most species of fish there was a high variability, meaning that any exponent (i.e., $0 \leq a < 1$) could be used to model growth without affecting the fit to the data significantly (when the other parameters $p$, $q$, $m_0$ were optimized). They explained this by data quality, as for

wild-caught fish and also for wildlife data there is always the problem of "haphazard" sampling, which may result in unreliable growth parameter estimates (*Wilson et al., 2015*).

Is this high variability for fish data still observed, if both exponents $(a, b)$ of the Bertalanffy–Pütter model are optimized? We explore the region of near-optimal exponent-pairs and hypothesize that the additional degree of freedom for the optimization of the exponent-pair (instead of the optimization of one exponent, only) would result in a significantly better fit of the optimal growth function and thus in a small region of near-optimality. Thereby, the term "near-optimal" (see Discussion) may be defined by different measures of the goodness of fit. Here we consider two such measures: One is the sum of squared errors (SSE), which comes from the most common approach to data fitting, the method of least squares; the other is the Akaike weight, which comes from the theory of model selection by means of the *Akaike (1974)* information criterion (AIC). For a discussion of alternative information measures, c.f. *Dziak et al. (2017)*.

## MATERIALS AND METHODS

### Study overview

We started with a literature search for mass-at-age data of fish. These data are exceptional, as most growth data for fish are length-at-age. In view of the computational complexity of optimizing the Bertalanffy–Pütter-model, we focus on one case study and identify optimal exponents for one fish data-set only.

Technically, given the data, we studied the function $SSE_{opt}(a, b)$, which for each exponent-pair $(a, b)$ identifies the minimal SSE that can be obtained optimizing the parameters $p, q, m_0$. As we aimed at evaluating and minimizing this target function $SSE_{opt}$ on a large grid of exponent-pairs, a fast and reliable optimization method was needed. We therefore started with several advanced general-purpose methods and a coarse grid in order to obtain a rough idea about the shape of $SSE_{opt}$ and the performance of the different methods. We then selected a method (interior point optimization) and applied it to a refined grid. Finally, we developed a custom-made method (based on simulated annealing) to identify the globally optimal exponent-pair (which no longer needed to be a grid point). As a complication, for the chosen data also the boundary diagonal $a = b$ of the parameter region (i.e., $0 \leq a < b$) needed to be considered, whence the same computations were repeated for this diagonal. This search of optimal parameters for Eqs. (1) and (2) used Mathematica 11.3.

For a given exponent-pair $(a, b)$, we then assessed the goodness of the fit of its optimal model curve to the data in relation to the globally optimal exponent pair (and its best-fitting model curve). We plotted the respective model curves, compared $SSE_{opt}(a, b)$ with the minimal value of $SSE_{opt}$, and used this information to compute also the respective values of AIC and the Akaike weights. We outline the details of our methodological approach below. There we also mention alternative approaches (i.e., different definitions of the target function). With respect to the results of this paper, we expected that different approaches might result in different optimal exponent-pairs (because then a different function is optimized). However, we expected that the general feature of the present optimization problem, such as the flatness of the target function in a large neighborhood of

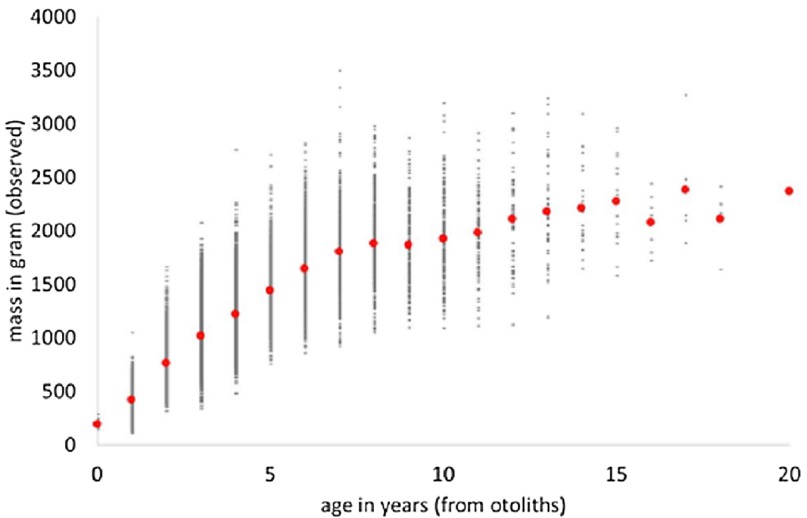

**Figure 1 Weight-at-age and average weight (red dots) of male Walleye from Lake Erie.**

the optimal exponent-pair (whence, e.g., elementary methods of optimization may fail) will persist independently of the other methodological details. We therefore selected a relatively elementary approach that nevertheless could be adapted to any of the other mentioned methodological variants.

## Data

We used "FSAdata WalleyeErie2" from *Ogle (2018)* about Walleye (*Sander vitreus*) from Lake Erie, USA, and retrieved a sub-sample (20,166 data-points) about male fish. The data informed about mass (in gram) and age (in years from otolits) of wild-caught fish. We applied MS Excel to retrieve the data and to pivot them (i.e., to identify average weights for the age classes). Figure 1 plots the data and the average weights.

There were few data about young fish (14 of age 0) and likewise few about older fish (22 with age 16–20 years), and none about fish with age 21–29 (maximal observed age reported in FishBase: *Froese & Pauly, 2018*). This may indicate gear bias (where small or large fish were not adequately sampled). In order to obtain more balanced class-sizes, smaller classes were merged; the outcome is Table 1, reporting of each class the average mass at the average age. Thus, 13 classes representing larger samples were evaluated instead of originally 20 age classes.

At first it may appear troubling to take more than 20,000 data points and then aggregate them to merely 13 mass-at-age classes. However, for data fitting it was the distance between the model curve and the average of each class that mattered. The distances between the average and the other class data could not be improved by a growth model. Further, in view of the large sample size it was reasonable to assume that average mass might be normally distributed (even if mass itself might follow another distribution), as in general average values are asymptotically normally distributed. However, we do not use this assumption.

**Table 1 Average weight-at-age (rounded) for male Walleye, based on ca. 20,000 age-weight data points (rounded to one decimal for the ease of presentation; the computations of the paper used data rounded to three decimals).**

| Age (years) | Weight (g) | Class size | Comment |
|---|---|---|---|
| 0 | 192.1 | 14 | |
| 1 | 423.7 | 4,009 | |
| 2 | 761.8 | 5,181 | |
| 3 | 1018.0 | 3,870 | |
| 4 | 1221.6 | 2,262 | |
| 5 | 1442.8 | 1,519 | |
| 6 | 1644.5 | 1,471 | |
| 7 | 1802.0 | 690 | |
| 8 | 1880.7 | 446 | |
| 9.5 | 1895.3 | 430 | classes 9 + 10 |
| 11 | 1982.6 | 105 | |
| 12.4 | 2140.4 | 104 | classes 12 + 13 |
| 15.3 | 2228.5 | 65 | classes 14–20 |

## General approach to data fitting

Numerical instability tends to impede fitting the generalized Bertalanffy model (i.e., optimization of $a$, $p$, $q$, $m_0$, when $b = 1$) to noisy data (*Shi et al., 2014*). Even for simple models (meaning: certain values for the exponents are assumed and the other three parameters are optimized) literature reported that optimization failed to converge for certain data sets (*Apostolidis & Stergiou, 2014*). One of the reasons was the use of parametrizations that require bounded growth functions (*Cailliet et al., 2006*), whereas not all data may support bounded growth. Another reason was the observation that even for simple models the problem of data fitting may overtask straightforward optimization routines. Clearly, with more parameters to optimize the problem of convergence becomes more demanding and also powerful methods slow down. In order to avoid running into numerical instability by the use of too many parameters, we used a grid search, where for each grid-point (exponent-pair $a$, $b$) we identified model parameters ($p$, $q$, $m_0$) that minimized the following function:

$$\text{SSE}_{\text{opt}}(a, b) = \min_{m_0, p, q} (\text{SSE}) \text{ for growth functions with exponents } a,\ b \tag{3}$$

There are various improvements of regression models, such as mixed-effect models to identify explanatory factors for growth (*Strathe et al., 2010*). However, such models require highly controlled experiments, whereas the present data are about wild-caught fish with unknown life history. In view of the difficulties with the convergence of optimization we did not consider more complex model assumptions, such as heteroscedastic growth that assumes a larger variance for a higher mass, or models that need additional parameters to distinguish different growth phases (*Manabe et al., 2018*). Further, in order to allow a meaningful visual comparison of the goodness fit of different model by an inspection of the plotted model curves, we did not use weighted sums of squared errors. Examples

for weights used in literature are the counts of fish at age and the reciprocals of the standard deviations of their masses. Otherwise, a curve whose plot appears to fit well may actually fit poorly, if it misses a heavily weighed data point. Thus, the purpose of optimization was the identification of a suitable growth curve for the considered species and not the identification of a growth curve that would minimize errors in relation to a given population. We did not simplify optimization by adding assumptions about parameter values, for example, eliminating two parameters from optimization by using a literature value for the initial condition $m_0$ (rather than optimizing it) and using a literature value for the asymptotic mass (defined below). In this case $SSE_{opt}(a, b)$ could have been computed very fast from the optimization of only one parameter, but at the cost of weakening the link to the data.

The use of grid-points helped to identify failures of optimization by a visual inspection (e.g., a grid-point with exceptionally high $SSE_{opt}$, when compared to neighboring grid-points). In order to do not miss the optimum, we used different approaches to data-fitting to identify and correct miscalculations. Thereby, computation time was an issue. For instance, commercially available software packages for fisheries management use powerful numerical methods to determine the model parameters even for the simple models (*Mildenberger, Taylor & Wolff, 2017*). These methods aim at optimizing one given model, where computing time is not an issue. Instead, here we aimed at optimizing a large number of models simultaneously in order to explore the function $SSE_{opt}$; that is, each grid point defined a model (defined from the exponent pair $a, b$) for which optimal parameters were identified. While for each grid-point $SSE_{opt}$ could be obtained fast, optimizing over the whole grid was time consuming. For example, covering the region $0 \leq a \leq 1$, $a < b \leq 3$ by a grid with neighboring points at distance 0.01 would define 25,250 grid points. For this grid, assuming six optimizations per minute would require 70 h of computing (CPU) time.

Optimization proceeded in three stages. First, $SSE_{opt}$ was computed on a coarse grid (step-size 0.1) to sketch the shape of $SSE_{opt}$ and locate a region of near-optimal exponents. This used methods of optimization that were fast, but not necessarily accurate. In the second stage, the computations were repeated with a finer grid (step-size 0.01) and using more accurate methods of optimization. These computations allowed to identify candidates for the optimum. In the final stage a search for the global optimum was performed, starting with these candidate points. The specific methods of optimization used in each step are explained below (c.f. the survey of *Cedersund et al., 2015*).

In order to speed up computations all approaches solved the differential equations (1) and (2) numerically (*Leader, 2004*). Using the analytic solutions of the differential equations (available in Mathematica) would make data fitting time consuming even for a given exponent pair. As the numerical methods used by Mathematica 11.3 work with high precision, this did not compromise the accuracy of optimization.

## Starting values for data fitting

For most iterative methods of optimization, reasonable starting values for the parameters are needed to ensure convergence of optimization. For instance, the starting value for the initial value $m_0$ was the first data point of Table 1.

For the other parameters, practitioners use various rules of thumb (*Carvalho & Santoro, 2007*), which utilize general considerations about the possible shape of the growth functions. For the typical solutions of (1) and (2) are increasing, bounded and sigmoidal. However, there are also non-sigmoidal solutions, for example, $a = 0$, and unbounded solutions, for example, $q = 0$ and $p > 0$. Initially the rate of growth increases, until the inception point is reached. Subsequently it decreases to zero in the limit, when the asymptotic mass $m_{max}$ is reached; there the right-hand side of (1) and (2), respectively, vanishes. For Eq. (1) with $a < b$ this results in the following equation:

$$m_{max} = \left(\frac{p}{q}\right)^{1/b-a} \tag{4}$$

To obtain a starting value $q_0$ for the parameter $q$, we assumed for the moment that the asymptotic mass would exceed the maximal observed mass by 20%, that is, we solved the equation $m_{max} = 1.2 \cdot \max(m)$ for $q$, referring to Eq. (4). This resulted in $q_0 = p_0/(1.2 \cdot \max(m))^{b-a}$, where $p_0$ was the starting value for $p$.

In order to obtain a starting value for $p$, we evaluated Eq. (1) approximately at $t = 0$, using for the right-hand side the above mentioned starting value $m_0$ for $m$ and $q_0$ for $q$. As approximate value for the derivative, $m'(0)$, we used the derivative at $t = 0$ of the quadratic interpolation polynomial (*Burden & Faires, 1993*) through the first three points listed in Table 1. This polynomial was an approximation for the growth function in the neighborhood of $t = 0$. Solving (1) for $p = p_0$ resulted in the following equation:

$$p_0 = \frac{m'(0) \cdot 1.2^b \cdot \max{(m)}^b}{1.2^b \cdot \max{(m)}^b \cdot m_0^a - 1.2^a \cdot \max{(m)}_a \cdot m_0^b} \tag{5}$$

These formulas defined starting values for $m_0$, $p$, and $q$. The formulas were problematic for exponents close to the diagonal, as the function $p_0$ tends to infinity in the limit $a \rightarrow b$. Therefore, for exponents $b = a + 0.01$ we used simulated annealing (see below) in case that optimization using these starting values did not converge.

**Preparatory screening**

$SSE_{opt}$ was computed for a coarse grid (distance 0.1 between adjacent points), using two general purpose methods for global optimization in parallel, simulated annealing and the Nelder–Mead amoeba method. Both methods are available for the Mathematica function NMinimize.

We used simulated annealing, as we expected it to produce reasonable results. It used random numbers as starting values (using multiple starting values) and then altered them by random fluctuations, accepting parameters with lower values of SSE, but also accepting with a certain probability (that became lower in subsequent iteration steps) parameters with a higher SSE to escape from suboptimal local extrema (*Vidal, 1993*). In order to ensure replicability, the default random seed 0 was used. Therefore, if SSE was optimized repeatedly for the same grid-point, the outcome remained the same.

We used the amoeba method because it is fast. Given the exponent-pair $a$, $b$, the method first evaluates four corners of a tetrahedron (simplex) in parameter space

(dimensions $m_0$, $p$, $q$) and successively applies reflections (moving the point with highest SSE through the opposite side of the tetrahedron to a point with perhaps lower SSE) and shrinking (zooming in to a local minimum point).

In order to avoid obviously meaningless parameter values, we added constraints to ensure an in relation to the data biologically reasonable initial value $m_0 > 10$ and positive parameters $p > q$.

### Semi-automated optimization

In order to employ also methods developed specifically for the least squares method, we used an alternative approach using the Mathematica function NonlinearModelFit. It implements the most common methods for nonlinear regression.

The optimization loop assumed a fixed value for $a$, whereas $b$ proceeded from $b = a + 0.01$ to $b = 2$ with step size 0.01. Further, for each exponent $a = n \cdot 0.01$ we plotted the hitherto obtained values of $\text{SSE}_{\text{opt}}(a, b)$. If the plot showed a U-shape, then we could identify a minimum of SSE on the line $a = n \cdot 0.01$, $b > a$; otherwise (human intervention) we added more values of $b$ to the loop until we could discern the U-shape. We thereby assumed that for still larger exponents $b$ the fit could only become worse. This assumption was corroborated by the initial screening.

The optimization started at $a = 0$, $b = 0.01$ with initial values for $m_0$, $p$, and $q$ explained above. For the subsequent computations, where $a$ was kept fixed and $b$ moved, the iterative optimization at the next $b$, namely at $b + 0.01$, started with the optimal parameters from the previous optimization (for $b$).

However, in order to ensure convergence (and an empirically meaningful outcome), we minimized SSE subject to certain constraints ($m_0 > 10$ and $q > 0$), whence many common methods from regression analysis (e.g., Levenberg–Marquardt algorithm) were not applicable. Instead, we used an interior point method. These methods (e.g., barrier methods initially developed in the 1960s) became popular in 1984, when an interior point method (*Karmakar, 1984*) solved linear optimization problems in polynomial time; *Forsgen, Gill & Wright (2002)* refer to the "interior point revolution." This setting was also advantageous for the present problem.

### Custom-made simulated annealing

Based on this preparatory work, we could evaluate $\text{SSE}_{\text{opt}}(a, b)$ for almost all grid points. In order to improve the estimates of SSE at the best fitting grid points and to move from there to the optimal exponent-pair (no longer a grid-point), we developed a custom-made approach of simulated annealing. We used the general purpose method of Mathematica in the preparatory screening, but its performance was suboptimal, whence modifications were needed to ensure convergence in reasonable time. The main difference to general purpose simulated annealing was the use of a (sort of) geometric Brownian motion. For each step, rather than adding a small random number to the parameters, they were multiplied by a random number, whence positive values were retained. The optimization used a loop with 500,000 steps: It started with the parameter values obtained from the preparatory optimization steps.

**Table 2 Optimal parameters for selected models.**

| Model | Comment* | a | b | $m_0$ | p | q | SSE |
|---|---|---|---|---|---|---|---|
| Bertalanffy | First (a, b given) | 2/3 | 1 | 203.8 | 11.2 | 0.86 | 23,709 |
| Logistic | First (a, b given) | 1 | 2 | 301.716 | 0.528051 | 0.000253611 | 72,283 |
| Optimal | Third (a optimized) | 0.686028 | = a | 175.67 | 21.3148 | 2.76054 | 21,286 |

Note:
* First and third refer to the initial and final rounds of optimization.

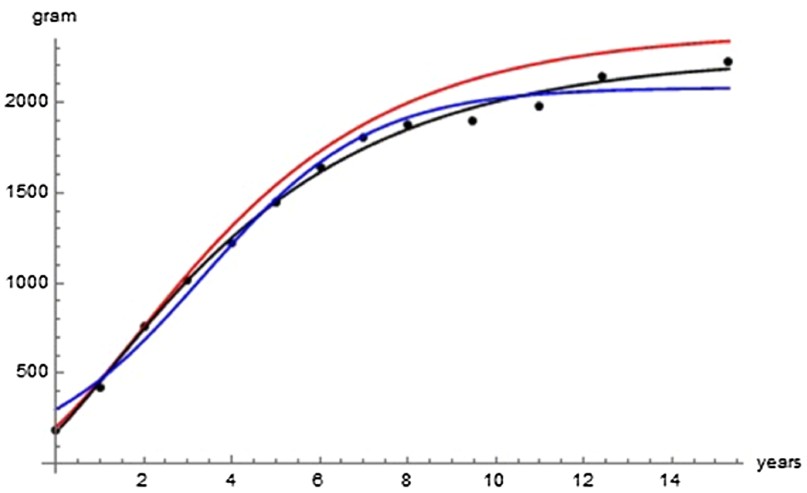

**Figure 2 Comparison with the data of the growth curve using the Bertalanffy exponent-pair (red), the logistic exponent pair (blue) and of the best fitting growth curve (black); parameter values as in Table 2.**

## RESULTS

Table 2 lists the optimal parameters for three exponent-pairs, Bertalanffy, logistic, and the optimal pair (of Gompertz-type), and Fig. 2 visualizes the model curves defined from these optimal parameters; all model curves remained reasonably close to the data. Failing optimizations often converged to a curve close to the mean value of the average masses. The following optimization aimed at finding improvements of $SSE_{opt}$ = 23,709 for the Bertalanffy-pair, which was obtained in the initial round of optimizations.

First round of optimization: We evaluated $SSE_{opt}$ at grid-points $0 \leq a \leq 1$ and $a < b \leq 1.5$, for growth functions (1) and at $0 \leq a = b \leq 1$ for (2). These grid-points were exponent-pairs at distance 0.1 between successive grid-points. For each grid-point the better of the outcomes from (general purpose) simulated annealing and from the amoeba method was used; $SSE_{opt}(0.7, 0.7)$ = 21,310 was optimal. However, the initial optimization became problematic for $b > 1.2$ and it did not allow to decide, if optimization would require a search in this problematic region. Further, it could not be decided if the optimum would be located on or above the diagonal.

Second round of optimization: We conducted a systematic search (semi-automated data fitting) confined to Eq. (1). It used a fine grid (distance 0.01 between successive exponent-pairs), aiming at identifying for each exponent $a$ with $0 \leq a \leq 1$ an exponent $b > a$ with minimal SSE. It was sufficient to screen exponents $b \leq 2$. The improved accuracy

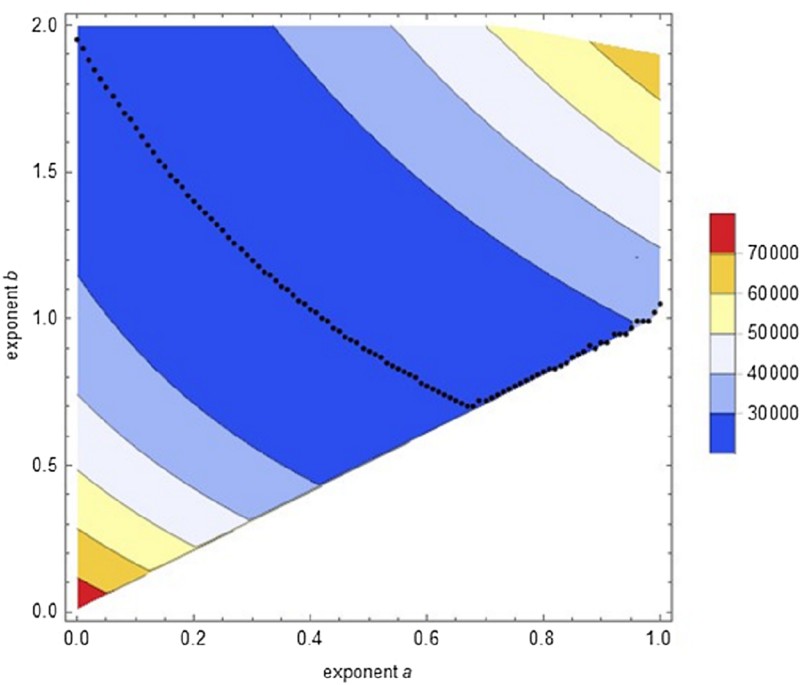

**Figure 3 Contour plot of the optimal SSE on a grid of exponent-pairs with distance 0.01 between adjacent points and for each exponent *a*, plot of the exponent-pair with smallest SSE (black dots).**

of this search was demonstrated for the Bertalanffy exponent-pair with a lower $SSE_{opt}(0.67, 1) = 23534.6$. Figure 3 plots the outcome from the optimization at 14,282 grid points (exponent-pairs). The black dots indicate, for each exponent *a*, for which exponent *b* the value of SSE was minimal. Thereby, $SSE_{opt}(0.67, 0.7) = 21287.1$ was the least observed optimized SSE for Eq. (1). This demonstrates that optimization showed the following pattern: For *a* = 0 the minimum SSE was reached close to *b* = 2. For the following values there was a distinct U-shape to be observed till *a* = 0.67. Finally, the optimum was attained close to the diagonal *a* = *b* (dots moving upwards), but the optimum value was increasing compared to the previous ones. This pattern supported the hypothesis that the optimal SSE would be attained within the (topological closure of the) search region. However, the computations did not allow to decide, whether the global minimum of SSE was attained for *b* > *a*, that is, for Eq. (1), or for *b* = *a*, that is, Eq. (2). Further, optimization proceeded smoothly till *a* = 0.7, but for larger exponents optimization became increasingly more difficult and fewer results could be accepted. In particular, grid points near the diagonal were problematic.

Third round of optimization: We tackled these issues using a global optimization. It started with the near-optimal parameters found previously. For Eq. (1), starting from *a* = 0.68 and *b* = 0.69, the least $SSE_{opt}(0.666703, 0.705181) = 21287.5$ was achieved. However, for Eq. (2), that is, on the diagonal *a* = *b*, a slightly better outcome $SSE_{opt}(0.686028, 0.686028) = 21286.4$ was obtained (parameters in Table 2). The custom-made method of simulated annealing improved insofar upon the same method as implemented by Mathematica (which was used in the initial step), as it was more accurate.

Further, despite the high number of computing steps its performance was more reliable (no unexpected computer crashes).

In summary, during the three rounds of optimization the fit achieved by the Bertalanffy exponent-pair ($a = 2/3$, $b = 1$ with $\mathrm{SSE_{opt}} = 23{,}709$) could be substantially improved. The first round identified a better exponent-pair ($a = b = 0.7$ with $\mathrm{SSE_{opt}} = 21{,}310$). The second round, using more accurate computations, found a still better exponent pair ($a = 0.67$, $b = 0.7$ with $\mathrm{SSE_{opt}} = 21287.1$). The final round converged to the minimal $\mathrm{SSE_{opt}} = 21286.4$ at $a = b = 0.686028$. Thus, by using different exponent-pairs and also by using more accurate optimization methods, $\mathrm{SSE_{opt}}$ could be reduced by 10% from the initial estimate using the von Bertalanffy pair.

## DISCUSSION

Optimization identified an exponent-pair that achieved a 10% reduction of $\mathrm{SSE_{opt}}$, when compared to the von Bertalanffy-pair. Was this reduction worth the efforts? The answer to this question depends on what notion of "near-optimality" is used (considered in this section) and for what purpose the model is needed (considered in the Conclusion).

An obvious definition of near-optimality would set a maximal percentage by which the optimal SSE may be exceeded. However, the appropriate percentage may vary with the data. Here, we explain a definition of near-optimality that refers to the Akaike weight; it therefore has the same meaning for all data. Specifically, we used an index $\mathrm{AIC_c}$ for small sample sizes (*Burnham & Anderson, 2002*; *Motulsky & Christopoulos, 2003*). $\mathrm{AIC_c}$ was defined from the least SSE for the model with exponent-pair $(a, b)$, that is, $SSE(\text{model}) = \mathrm{SSE_{opt}}(a, b)$, from the number $N = 13$ of data-points (size of Table 1 rather than the number of fish), and from the number $K$ of optimized parameters:

$$\mathrm{AIC_c}(\text{model}) = N\ln\left(\frac{\mathrm{SSE}(\text{model})}{N}\right) + 2 \cdot K + \frac{2 \cdot K \cdot (K+1)}{N - K - 1} \tag{6}$$

$$\mathrm{prob}(\text{model}) = \frac{e^{-\Delta/2}}{1 + e^{-\Delta/2}} \text{ , where } D = \mathrm{AIC}(\text{model}) - \mathrm{AIC}(\text{best fitting model}) > 0 \tag{7}$$

The Akaike weight prob compares a model with the best fitting model in terms of the least $\mathrm{AIC_c}$: Its Akaike weight prob(model) is the probability that this model is true (assuming that one of the two models is true); the maximal Akaike weight is 50%. This interpretation is based on the assumption of normally distributed errors. As the data were average values of large samples, this assumption was justified. However, the Akaike weight may also be interpreted as just another measure of the goodness of fit to the data; see below. Such an interpretation does not need the assumption of a normal distribution.

Technically, the application of the above criteria requires that two distinctions are made: First, the differential equations (1) and (2) that set the general framework for this study need to be distinguished from the different growth models that may or may not assume specific values for the exponent-pair. Thereby, each grid point defined a concrete model of type (1) with an assumed exponent-pair $(a, b)$; for example, logistic model with

$(a, b) = (1, 2)$. The (other) model parameters $(m_0, p, q)$ were optimized (data fitting). However, the third round of optimization in addition sought for optimal exponents, referring to the general Bertalanffy–Pütter model and the general Gompertz model, respectively. Thereby, the AIC of models with assumed exponent-pairs was computed with $K = 4$ (as implicitly also SSE was optimized). The AIC of the general Bertalanffy–Pütter model and the general Gompertz model was computed with $K = 6$ and $K = 5$, respectively, as also the exponents were optimized. Owing to this penalty for additional parameters, the best fitting model in terms of the least SSE could have a higher (worse) AIC than other models. Second, we interpreted the Akaike weights in two ways. If the AIC was computed with the above explained correct number of parameters, the Akaike weights might be interpreted in the usual way as probabilities about the truth of a model. However, we also used the Akaike weights with an incorrect number of parameters, assuming $K = 4$ for all models; that is, also the models with optimized exponents were treated as if these exponents were given in advance. For this application, the Akaike weight was merely a measure of the good fit (low SSE) that was comparable across different data-sets, but not a probability of truth.

We use the second interpretation to define acceptability and near-optimality (*Renner-Martin et al., 2018*): A model defined from an assumed exponent-pair $(a, b)$ has an acceptable fit, if in comparison to the optimal exponent-pair its Akaike weight is 2.5% or higher (i.e., the lowest 5% of Akaike weights are deemed as inacceptable), whereby all Akaike weights are computed with $K = 4$ (assuming that the optimal exponent-pair was given in advance). The exponent-pairs with an acceptable fit define the region of near-optimality. Using some algebra, this definition is equivalent to the following condition in terms of SSE, $N$ and $t$ (the above threshold), which defines acceptability by a maximal percentage (dependent on $N$), by which the optimal SSE may be exceeded (e.g., for $N = 13$ and $t = 0.025 = 2.5\%$, an excess of 75.7% is acceptable):

$$\frac{\text{SSE(model)}}{\text{SSE(best fitting model)}} < 1.757 = \left(\frac{1}{t} - 1\right)^{2/N} \tag{8}$$

Figure 4 (all Akaike weights computed with $K = 4$) shows that amongst generalized von Bertalanffy models (defined by exponent pairs with $b = 1$), the comparison with the best-fitting model did affect the Akaike weights only slightly. For instance, for the Bertalanffy pair the Akaike weight was reduced from 36% (comparison with the optimal exponent $a$, assuming $b = 1$) to 34% (comparison with the best-fitting exponent-pair). For lower Akaike weights the reduction was even smaller, whence the Akaike weights could not be pushed below the 2.5% threshold. Thus, despite the comparison with the overall optimal Bertalanffy–Pütter model, for the class of generalized von Bertalanffy models ($b = 1$) all exponents $0 \leq a < 1$ were acceptable.

Figure 5 illustrates how this variability extended into two dimensions (the dimensions referring to the number of considered exponents). The green area represents exponent-pairs, whose AIC was below the correct AIC of the best-fitting model. Thereby, AIC for given exponent-pairs was computed with $K = 4$, while the AIC for

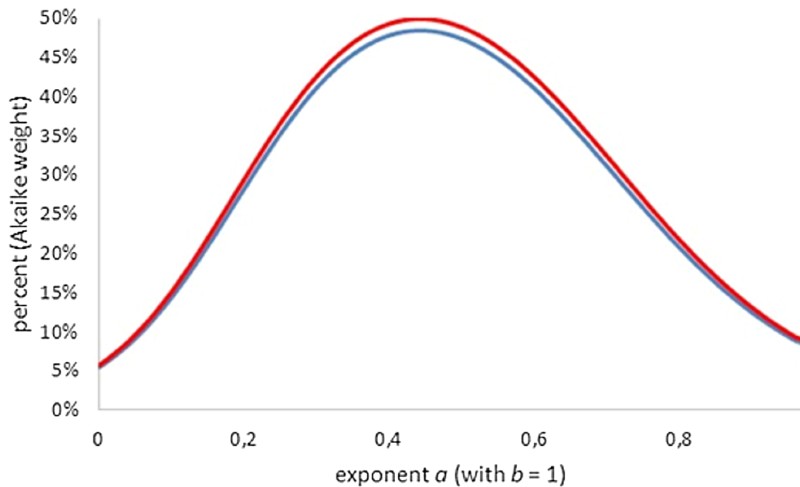

**Figure 4** Plot of the Akaike weights for exponent-pairs with $b = 1$, using the least AIC amongst generalized Bertalanffy-models (red) and the least AIC amongst all considered models (blue); all AICs using $K = 4$.

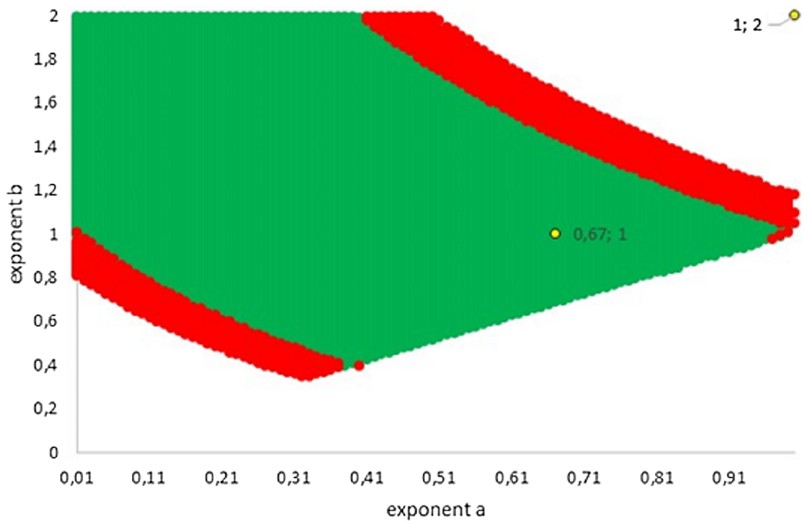

**Figure 5** Plot of the grid points $a < b$ with AIC below AIC of the best fitting model (green; the AIC of the best fitting model was higher due to the penalty for an additional parameter) and with acceptable fit (red). The Bertalanffy and the logistic exponent-pairs are displayed in yellow.

best fitting Gompertz-type model was computed with $K = 5$, whence there was a penalty. The red area represents additional exponent-pairs, whose fit was deemed as acceptable in the meaning above (Akaike weight of 2.5% or higher, using $K = 4$ also for the best fitting model). The red area was bounded, meaning that in two dimensions not all exponent-pairs were acceptable.

The following examples illustrate these concepts. In Fig. 2, the best fit in terms of SSE was achieved by the optimal exponent-pair, followed by the von Bertalanffy-pair, while logistic growth had the poorest fit. However, owing to the penalty in the definition of AIC for using more parameters, the von Bertalanffy exponent-pair was in the green region of
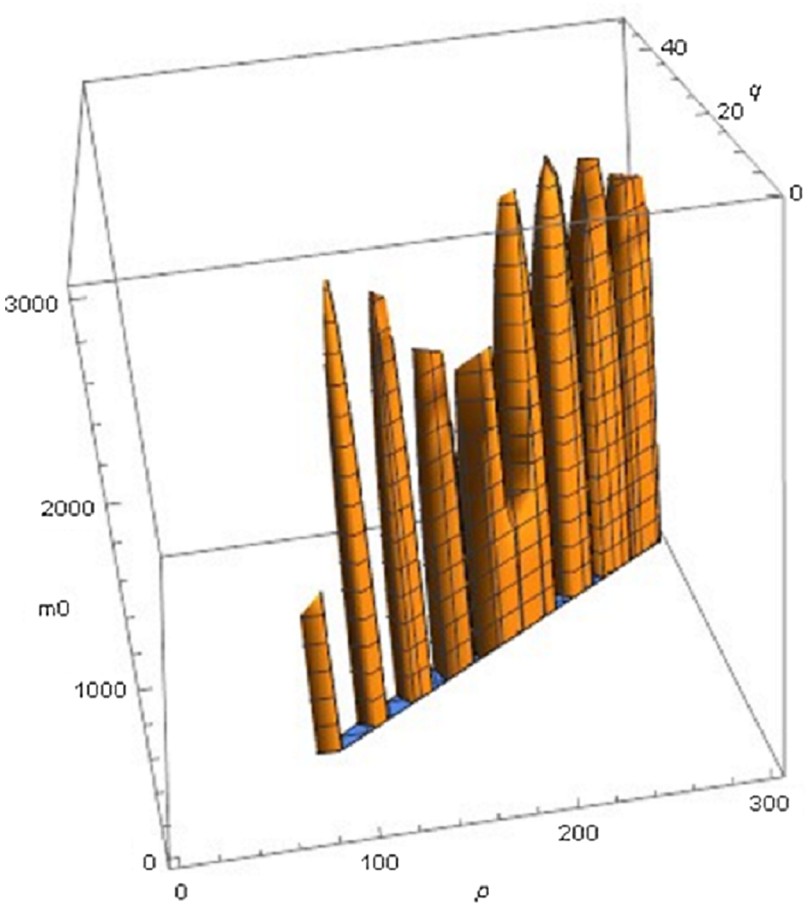

**Figure 6 Plot of part of the region of exponents $m\_0$ $p$, $q$ for model (2) with the optimal exponent $a = 0.686028$, where SSE does not exceed $10^7$.**

Fig. 5. Therefore, when choosing between the von Bertalanffy and the best fitting exponent-pair, the criterion of parsimony would recommend to select the former one. The logistic exponent-pair was outside the red or green regions of Fig. 5, whence this fit was deemed as not acceptable, although in Fig. 2 it still appeared to be reasonable. In summary, when comparing these exponent pairs, the von Bertalanffy-pair would be selected as parsimonious; the logistic pair would be refuted due to its poor fit; and the optimal pair would be refuted, as its 10% reduction of $SSE_{opt}$ (in comparison to the von Bertalanffy exponent-pair) did not justify the optimization of an additional parameter. Figure 6 indicates that Eqs. (1) and (2) may indeed result in overfit due to the optimization of too many parameters. Using model (2) together with the optimal exponent, it plots the region of the "other parameters" ($m_0$, $p$, $q$), where SSE was bounded by $10^7$ (ca. 500 times the least SSE). Despite this large SSE, the region was extremely thin, suggesting some relation between the parameters. This indicates that a subclass of the Bertalanffy–Pütter model using fewer parameters may provide the same fit and therefore suffice for the modeling of growth. There remains the problem to find such a subclass that in addition is empirically meaningful.

## CONCLUSION

The paper conducted a case study about Bertalanffy–Pütter exponent-pairs ($a$, $b$) for fish. It was based on mass-at-age data of Walleye (*S. vitreus*). Comparing the von Bertalanffy exponent-pair model with the general Bertalanffy–Pütter and Gompertz models, the general models reduced bias in growth estimates (SSE) but increased model complexity. However, there was a large region of near-optimal exponent-pairs, amongst them the von Bertalanffy exponent-pair. Therefore, parsimonious model selection (AIC) confirmed the established practice to describe growth in term of the Bertalanffy models (VBGF for length, the von Bertalanffy exponent-pair for mass). However, there are purposes where the predictive power of models and the good fit of the model curve to the data is more important than simplicity. For example, if a model is used to justify policy recommendations on the basis of certain data, such as size-based catch-limitations in fishery stock assessments, it is of crucial importance that the model curve fits well to the data. Further, even in cases where the Bertalanffy-exponent pair is the most parsimonious model, it may not be true, as biological arguments (e.g., about metabolism) may support different exponent-pairs (e.g., the models mentioned in the introduction). Thus, *Pauly (1981)* recommended to use the class of generalized von Bertalanffy models ($b = 1$ and the exponent $a$ is free). We go one step further and recommend the use of the Bertalanffy–Pütter model (and of its limit case, the generalized Gompertz model) in natural resources management and other contexts that rely on careful quantitative assessments.

With respect to future research, we speculate that for the best fitting parameters there may exist additional relations, whence optimization might be further constrained by some functional relationship between the parameters. This would define a subclass of Bertalanffy–Pütter models with optimal or near-optimal fits. In order to identify it, we suggest to evaluate the optimal exponent-pairs for different data and species and search for a biologically meaningful pattern of these exponent-pairs. However, for this task it may be necessary to use different target functions for the evaluation of the goodness of fit.

## ACKNOWLEDGEMENTS

The authors appreciate the insightful comments by two reviewers that improved this paper substantially.

### Funding

Katharina Renner-Martin was supported by a grant from the University of Natural Resources and Life Sciences, Vienna. The funders had no role in study design, data collection and analysis, decision to publish, or preparation of the manuscript.

### Grant Disclosures

The following grant information was disclosed by the authors:
University of Natural Resources and Life Sciences, Vienna.
## Competing Interests

The authors declare that they have no competing interests.

## Author Contributions

- Katharina Renner-Martin analyzed the data, contributed reagents/materials/analysis tools, prepared figures and/or tables, authored or reviewed drafts of the paper, approved the final draft.
- Norbert Brunner analyzed the data, contributed reagents/materials/analysis tools, prepared figures and/or tables, authored or reviewed drafts of the paper, approved the final draft.
- Manfred Kühleitner analyzed the data, contributed reagents/materials/analysis tools, prepared figures and/or tables, authored or reviewed drafts of the paper, approved the final draft.
- Werner-Georg Nowak analyzed the data, contributed reagents/materials/analysis tools, prepared figures and/or tables, authored or reviewed drafts of the paper, approved the final draft.
- Klaus Scheicher analyzed the data, contributed reagents/materials/analysis tools, prepared figures and/or tables, authored or reviewed drafts of the paper, approved the final draft.

## Data Availability

GibHub: https://raw.githubusercontent.com/droglenc/FSAdata/master/data-raw/WalleyeErie2.csv.

The raw data is included in Table 1 and the Supplemental Files.

## Supplemental Information

Supplemental information for this article can be found online at http://dx.doi.org/10.7717/peerj.5973#supplemental-information.

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
