# Peer review of "Optimal and near-optimal exponent-pairs for the Bertalanffy-Pütter growth model"

_PeerJ, doi:10.7717/peerj.5973_

## Round 0.1 · original submission · Minor Revisions

The reviewers commented positively on the experimental design and extensive data set. However, concerns were raised about the clarity and style of writing. The authors should use the detailed feedback to strengthen the manuscript before it can be accepted for publication.

Reviewer 1 ·

Basic reporting

English language and style are fine/minor spell check required:
1) Review the use of colons, commas, periods, semi-colons for the equations in the lines 44,61,106,169,179,291 and 292.
2) line 169: improve writing, specifically remove the text ", eq. (1)".


Review the line 62, the definition of hypergeometric function according to Seaborn (Hypergeometric functions and their applications, V8, pp 50) it is a solution of a second-order linear ordinary differential equation (ODE). Thus, the proposed model of first order ODE does not fit. However, at work (Ohnishi, Yamakawa & Akamine, 2014) finds a relationship with incomplete beta function which is a special type of hypergeometric function. In this relation the solution x obeys the form B (x; a, 0) -B (x0; a, 0) = kt. Note that x is not a function of incomplete beta function.

Seaborn, J. B. (2013). Hypergeometric functions and their applications (Vol. 8). Springer Science & Business Media.

Experimental design

I commend the authors for their extensive data set. In addition, the manuscript is clearly written in professional, unambiguous language. The statistical analysis is very good and well summarized. However, the authors suggest in the title, abstract, introduction and conclusion that only SSE and AIC are sufficient to optimize the parameters of the Pütter-Bertalanffy equation. However, they use very advanced tools when compared to SSE and AIC such as Nelder-Mead amoeba and Simulated Annealing methods. This mathematics appears extensively in the sections Preparatory screening, Semi-automated optimization and Custom-made simulated annealing. I recommend reviewing the abstract and conclusion which should be improved upon before Acceptance. Summarizing, it is not possible only with SSE and AIC to get article completion.

Validity of the findings

The paper is well-written and correct. Many new results are derived, which are original and worth for publication. The masses collected (20167 fish) follow a normal distribution but it is not indicated if this choice was intentional.

Additional comments

The paper has many interesting results and can be published if the authors address after minor revision.

·

Basic reporting

The article is structured well. The literature referenced is appropriate, the results are relevant and test some hypotheses, but these hypotheses could be clearer (see below). The figures and table are clear and useful.

I do think the paper suffers from a few key problems that make things unclear and there are some English language problems. One of the issues I found with the paper was flow: the authors use a lot or 'passive tense' rather than 'active tense'. Hence, I do recommend some major re-writes to increase clarity (but I don't recommend any changes to the study design).

Examples here include:

Line 27: 'it was conjectured' rather than 'We hypothesized'. It is okay to use the first person, this will make things more clear and direct with less chance to confuse readers. It will also reduce word usage and avoid wanting to invert sentence clauses.

Line 100: 'As Shi et al. (2014) observed, already for the generalized Bertalanffy model (i.e. b = 1, a, p, q, m0 are optimized) data fitting was impeded by numerical instability.' I suggest re-writing this sentence to say: Numerical instability tends to imped fitting the generalized Bertalanffy model (i.e. optimization a, p, q, m0 when b = 1) to noisy data (Shi et al. 2014).

Line 104: change 'the paper' to either first or third person throughout the whole paper. I recommend first person and just say something like: "we used a grid search to estimate growth model exponents.'

Furthermore, the paper has some odd sentences that appear incomplete. For example, line 26 in the abstract:

"Thereby, data fitting used the method of least squares, minimizing the sum of squared
27 errors (SSE)." This sentence has some uncertainty what point the "thereby" is following up. A second problem is that the objective of the sentence "data fitting used the method of least squares" is unknown because the key points or knowledge gaps being tested in the paper have yet to be introduced. I would re-write these portions of the abstract to introduce the topic, then the knowledge gaps, then the objectives in the paper. And its okay (and clearer) to be direct.

Another challenge with passive tense is that the major conclusions appear soft and vague rather than clear and direct.

Line 33: " SSE is used, when predictive power is needed alone, and AIC is used, when simplicity of the model and explanatory power are needed." This sentence needs a re-write to be direct and active. Tell the reader your conclusions/recommendations such as: 'We recommend estimating the general VBGF in cases when predictive power and bias reduction are preferred (such as in fishery stock assessments), although we note the special case VBGF remains a parsimonious model in many cases.' This is a really key result but the sentence structure masks the finding.

Another key problem I found was that I didn't fully understand the key objectives of the paper until the very end. Neither the Abstract nor Introduction clearly sets up the main objectives.

The objective is loosely stated on line 75, but I don't think this does a suitable job describing what is being done. What are the two dimensions? What variability is referred to? Are you more interested in reducing unexplained variability? Is it variability in growth parameter estimates themselves?

Also, some paragraphs need to be merged or separated. For example, line 257 and line 263 are new paragraphs but their topic sentences refer to results in a previous paragraph. Either merge these points into a single paragraph or make the paragraphs more independent of previous paragraphs.

Experimental design

I found the experimental design very competent. The presentation of the model development and the analyses conducted is clear and expert. I congratulate the authors on this!

The research questions are defined, but not always very clear or meaningful. The authors can see my above comments about the language problems. I think the authors need to use active tense and state quite directly what the knowledge gaps are, what the objectives are, and what the hypotheses were (conjecture is too weak of a word, in my opinion). These are all hinted at in the introduction, but not clearly and I think the average reader may get lost on the points in the Introduction and Methods which set up the study design.

For example, Line 257 suggests there was a hypothesis that a and b would be correlated, and that optimal SSE would be found along the diagonal of the a -> b grid search. Where was this hypothesis stated? The methods hint at it, but make this more clear.

I would recommend a single paragraph early in the Methods which is a "Study Overview" where the authors use very general language to describe the problems and general solutions/approaches to solve these problems. By the time we get to the other sections in the Methods, the authors expertise is obvious and the details are useful for replication by other expert readers.

Validity of the findings

At times reading the Discussion and Conclusion, I was still unsure what the take home conclusions ought to be. Should we use the general VBGF or should we still use the special case VBGF. These nuances are described, but I think the authors should be clear about when it works and when it doesn't. I think this is mostly a language issue though, and so some revisions in the Discussion and Conclusions should solve this problem. Without a suitable revision on these problems, I think the main findings will remain elusive for a lot of readers.

The grid search estimation of the general exponents reduced bias via SSE but was not the most parsimonious model via AIC. I like the use of the two model performance criteria, I wonder if these benefits could be expanded further in the Discussion. What about bias reductions in fishery stock assessments?

The lead sentence in the Discussion poses a question, but it is so specific to the results that a general reader doesn't know what to take home. What 10% reduction? What SSE? What two exponents?

Instead, I recommend simply saying a more general conclusion like: "Estimating the general VBGF reduced bias in growth estimates but increased model complexity. We are thus left with the remaining question: is this improved performance worth additional complexity? According to parsimonious model selection, like AIC, the answer is no. However, this bias reduction may be worth the tradeoff with complexity in some situations where predictive power is solely preferred, like natural resource management, which relies on careful quantitative assessments to recommend policies for sustainable resource usage." Obviously this can be changed up, but I think it makes the key points and remaining questions more clear to the average reader (which may be fishery/resource managers and practitioners).

---

## Round 0.2 · accepted · Accept

The authors have adequately addressed all comments.

# Reviewer 1 ·

Basic reporting

no comment

Experimental design

no comment

Validity of the findings

no comment

Additional comments

This is a very good paper.

·

Basic reporting

The authors were able to make my recommended revisions on sentence structure, and I found the reporting in the paper improved and sufficient. Thanks for making those changes!

Experimental design

I really enjoy the way the model design for the paper. I think the revisions helped to clarify what the overall questions were, to which the design then helps address and find answers for.

Validity of the findings

I think the findings are quite robust, as evidenced by the statistical design the authors used. I think the findings are somewhat expected (that the special case VBGF is generally the most parsimonious, but not always the best for prediction).

The discussion is useful to couch these findings more generally in the fisheries literature.

Additional comments

Good job on the revisions, I was very satisfied with the new changes in the discussion and the framing of the objectives/hypotheses.